# A computational model for bacteriophage φX174 gene expression

Alexis M. Hill[1]ʘ, Tanvi A. Ingle[1]ʘ¤, Claus O. Wilke[1]*

**1** Department of Integrative Biology, The University of Texas at Austin, Austin, TX, United States of America

ʘ These authors contributed equally to this work.
¤ Current address: University of Texas Southwestern Medical Center, Dallas, TX, United States of America
* wilke@austin.utexas.edu

**Data Availability Statement:** Our simulation and analysis scripts are available at: https://github.com/alexismhill3/phix174-simulation.

**Funding:** This work was supported by a National Institutes of Health grant R01 GM088344 and by the Jane and Roland Blumberg Centennial

## Abstract

Bacteriophage φX174 has been widely used as a model organism to study fundamental processes in molecular biology. However, several aspects of φX174 gene regulation are not fully resolved. Here we construct a computational model for φX174 and use the model to study gene regulation during the phage infection cycle. We estimate the relative strengths of transcription regulatory elements (promoters and terminators) by fitting the model to transcriptomics data. We show that the specific arrangement of a promoter followed immediately by a terminator, which occurs naturally in the φX174 genome, poses a parameter identifiability problem for the model, since the activity of one element can be partially compensated for by the other. We also simulate φX174 gene expression with two additional, putative transcription regulatory elements that have been proposed in prior studies. We find that the activities of these putative elements are estimated to be weak, and that variation in φX174 transcript abundances can be adequately explained without them. Overall, our work demonstrates that φX174 gene regulation is well described by the canonical set of promoters and terminators widely used in the literature.

## Introduction

φX174 is a single-stranded circular bacteriophage from the family *microvirdae*. It has a small, 5386 nucleotide genome comprised of 11 genes, which are organized into two distinct functional clusters within the genome [1, 2]. The structural genes (J, F, G, and H) form one contiguous genomic region, and encode proteins that enclose and protect the phage genome. The remaining seven genes encode proteins that facilitate viral propagation, for example by disrupting host cell replication or catalyzing cell lysis. φX174 gene regulation is relatively straightforward and depends largely on the joint activities of its promoters and terminators [3, 4]. Notably, the φX174 genome is also extremely compact, in that half of its genes overlap with at least one other coding region. Due to its tractable size and unusual genomic architecture, φX174 has served as an important model organism across the fields of structural and synthetic biology, and has been the subject of numerous efforts to re-organize, re-code, or modularize its genome [5–7].

Professorship in Molecular Evolution and the Dwight W. and Blanche Faye Reeder Centennial Fellowship in Systematic and Evolutionary Biology at The University of Texas at Austin. The funders had no role in study design, data collection and analysis, decision to publish, or preparation of the manuscript.

**Competing interests:** The authors have declared that no competing interests exist.

Characterization of $\phi$X174 transcription regulation has been ongoing since even before it was first sequenced nearly 50 years ago [2], with subsequent refinements being made as more sophisticated molecular technologies become available. First, the locations of $\phi$X174 promoters and terminators were mapped for *in vitro* [8, 9] and *in vivo* [10] transcription regulation using a combination of selective oligonucleotide initiation and DNA hybridization. Follow-up studies verified the initial mapping using sequence analysis [11]. For the promoters, *in vitro* kinetic activities were also established [4]. However, these assays can be difficult to set up in such a way that they accurately capture *in vivo* activity. For example, measurements for $\phi$X174 promoter activities can vary by several orders of magnitude, depending on how much original sequence context is retained around the cloned promoters [4]. Furthermore, the $\phi$X174 regulatory map itself may be incomplete. For example, Logel and Jaschke [12] proposed that two additional regulatory elements (one promoter and one terminator) may play a role in $\phi$X174 gene expression regulation, based on novel high-resolution RNA-seq measurements of phage transcription. Although $\phi$X174 transcription has been well studied, additional work may be needed before transcription regulation for $\phi$X174 can be fully resolved.

Here, we investigate $\phi$X174 gene expression regulation using mechanistic, computational simulations. We infer kinetic values for $\phi$X174 promoters and terminators by fitting simulations to transcription data [12, 13], and we assess the effect of two putative regulatory elements [12] on global $\phi$X174 transcription patterns. We also use simulation to explore the consequences of $\phi$X174 genome decompression [5]. Overall, we find that rate constants for regulatory elements can be reliably identified and are broadly consistent with prior kinetic measurements. These observations suggest that the current understanding of $\phi$X174 transcription regulation is reasonably complete. However, we also observe that parameter estimation is somewhat ambiguous in the case of a promoter immediately followed by a terminator, an arrangement that occurs once in the $\phi$X174 genome. In this case, the terminator partially compensates for the promoter, and the model cannot distinguish between a strong promoter followed by a strong terminator or a weak promoter followed by a weak terminator. Our approach is general and could be applied to study gene regulation in other organisms where transcription control is less well understood.

## Results

### A model for $\phi$X174 gene expression

In order to simulate phiX174 gene expression regulation, we built a model for bacteriophage $\phi$X174 using a customizable stochastic simulation framework that was previously used to model bacteriophage T7 infection dynamics [14, 15]. The $\phi$X174 infection cycle begins with injection of the single-stranded phage DNA into the host cell [16], followed by synthesis of the complementary DNA strand, which serves as the mRNA template. Our simulations begin with transcription and assume that the fully double-stranded phage DNA is already synthesized. We modeled the phage genome as a linear molecule with a start site 63 nucleotides upstream of gene A (Fig 1A). To simulate circular genome transcription, polymerases that reach the end of the genome without terminating automatically begin another round of transcription starting from the beginning of the genome (Fig 1B). We modeled background host-cell expression by simulating transcription of a single ORF of average length for *E. coli*. During the simulation, *E. coli* gene expression competes with the phage for cellular resources (polymerases, ribosomes). Finally, we defined initial resource availability and transcription/translation kinetics using parameters derived from the prior bacteriophage T7 gene expression model [15] (see also Materials and methods).

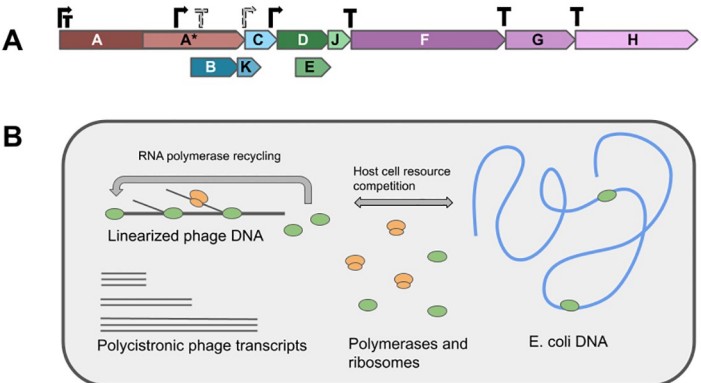

**Fig 1. Simulation overview.** A: The $\phi$X174 genome is linearized starting from gene A. Reading frames for genes A*, B, K and E overlap with at least one other gene. Canonical promoters (solid black symbols) and the putative promoter and terminator (gray with a dashed boarders) are shown directly above the gene diagram. B: $\phi$X174 gene expression is simulated for a single *E. coli* cell. Transcription and translation of the phage genome is modeled mechanistically, with explicit tracking of ribosomes and polymerases as they traverse DNA and RNA polymers. *E. coli* gene expression is modeled as a set of reactions that consume/produce resources utilized by the mechanistic simulations.

## Fitting the model to qPCR data

Since exact values for promoter and terminator strengths were not known *a priori*, we estimated these values by fitting the simulations to $\phi$X174 transcription data. For the fitting procedure, promoters and terminators were initialized arbitrarily and then fit using a simple iterative optimization procedure that adjusts one randomly selected element at a time. During model optimization, new parameter values were retained only if they decreased the error between the steady-state simulation output and the target transcription pattern. Specifically, we calculated the root mean squared error (RMSE) between final simulated transcript abundances and measurements of $\phi$X174 transcription taken several minutes into the infection cycle.

As a proof of concept, we first fit our model to low-resolution qPCR data of $\phi$X174 [13]. The data set consists of transcript abundances mapped to six locations on the $\phi$X174 genome. We seeded 20 independent simulations and fit each to these measurements (Fig 2A). After 4000 generations, all 20 simulations converged with RMSE's <5.5 (Fig 2B, S1 Fig) and output that qualitatively matched qPCR measurements. We manually inspected the simulated transcription patterns and RMSE values and defined 2.75 as the cutoff RMSE score for further analysis; 14 out of the 20 fitted models met this criterion.

Next, we compared parameter estimates from the fitted models to empirical measurements of $\phi$X174 promoter and terminator activities. Fig 3A and 3B show estimates for the three canonical $\phi$X174 promoters and four terminators, respectively, from fitted models with RMSE's below the cut-off value. $\Phi$X174 terminator efficiencies have been measured to be about 40–60% for terminators $T_J$, $T_F$, and $T_G$, and 90% for terminator $T_H$ [17]. Using mean parameter values from our fitted models, we estimated an efficiency of 0.93 for terminator $T_H$, and efficiencies of 0.34, 0.51, and 0.78 for terminators $T_J$, $T_F$, and $T_G$, respectively. Thus, we concluded that our terminator estimates were reasonably consistent with experimental values.

Using empirical measurements as a baseline for promoter comparison is more challenging, since *in vivo* kinetic characterizations have found different relationships for pA, pB, and pD [4, 18]. For example, Sorensen et al. [4] measured an 18-fold difference between pA and pB (with pA > pB) when 90 bp of sequence context was included around cloned $\phi$X174

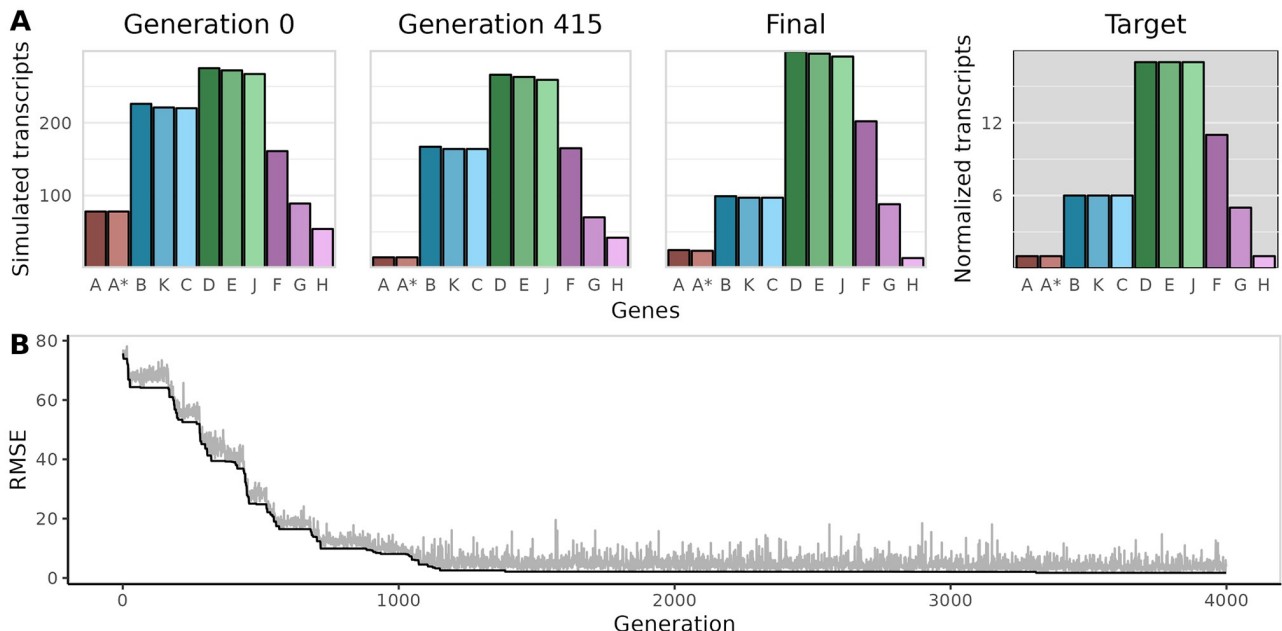

**Fig 2. φX174 models converge to qPCR measurements.** A: Time course for one representative model, out of the 20 that were fit to data. Target transcript abundances were measured using qPCR [13] and are shown normalized to gene A. The time course shows final simulated transcript abundances for the initial simulation (generation 0), a simulation from the middle of model optimization (generation 415), and the final optimized simulation that minimizes the RMSE. Output from the final simulation is qualitatively similar to the target data. B: The RMSE declines with increasing generations as fitted promoter and terminator strengths produce more accurate transcription patterns.

promoters. (There was no detectable transcription signal from cloned pD sequences.) In contrast, activities for all three major promoters were approximately the same (S1 Table) for constructs that contained a smaller amount of the original flanking sequence context. In our fitted model, the promoter strengths necessary to achieve target transcription patterns were intermediate between these results; the largest difference between promoters (pB and pD) was about 10 fold, with pD $\gg$ pA > pB.

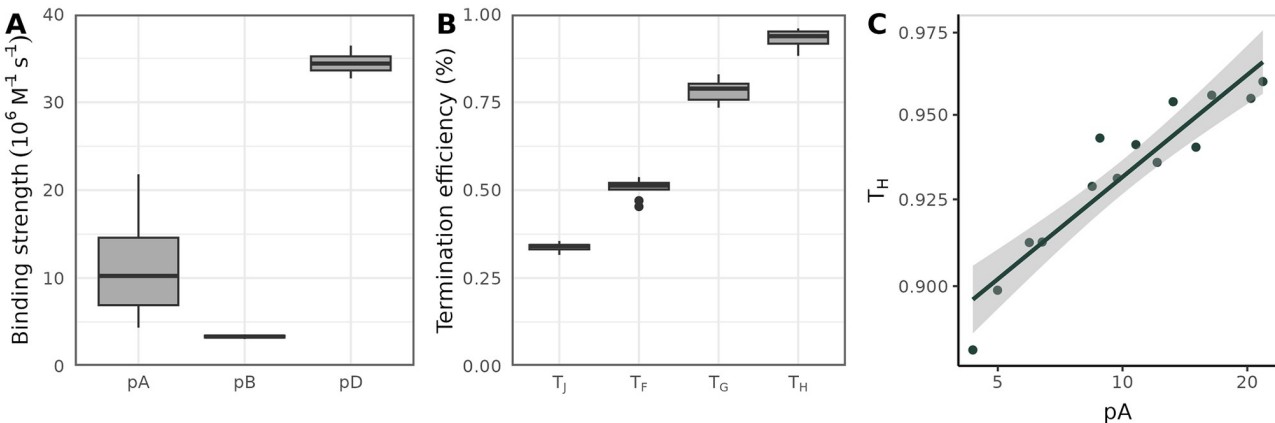

**Fig 3. Promoter and terminator strengths estimated from qPCR data.** Promoter strength distributions (A) and terminator strength distributions (B) from models with a final RMSE < = 3. The qPCR data used to fit all models was originally reported by Zhao et al. [13]. (C) Relationship between estimates for promoter pA and terminator $T_H$; each point represents a pA–$T_H$ pair from one fit model.

For most of the fitted parameters, variation between each simulation is low, suggesting that a single set of parameters minimizes model error. The exception is promoter A, for which there is a relatively wide range of values that provide equally good fits to the data (Fig 3A). In the $\phi$X174 genome, promoter A is situated less than 100 bp upstream of the strongest $\phi$X174 terminator, $T_H$. As a consequence, the majority of transcriptional current originating from pA is cut-off by termination at the $T_H$ site. Given the proximity of the two elements, we reasoned that their activities were likely coupled in the simulations. In other words, $T_H$ compensates for pA (or vice versa), such that a range of $T_H$–pA values is capable of producing the same gene expression pattern. There is a strong positive relationship between estimates for pA and $T_H$ (Fig 3C). Specifically, pA appears to be extremely sensitive to small deviations in the efficiency of $T_H$.

If $\phi$X174 gene expression is optimal (in the sense that it maximizes phage fitness) then a compensatory relationship between pA and $T_H$ could be beneficial, since it would provide some additional flexibility for maintaining optimal expression. For example, a mutation weakening the affinity of pA for *E. coli* polymerases could be rescued by a subsequent decrease in $T_H$ stability, in addition to reversion of the original mutation. Alternatively, it could be that $\phi$X174 biology requires the specific ordering of pA followed by $T_H$, and that coupling between the two elements is simply a consequence of this requirement. When we simulated $\phi$X174 gene expression with the order of pA and $T_H$ reversed (effectively de-coupling their activities, since $T_H$ stops most transcriptional current) the promoter A strength needed to maintain the same level of expression (keeping all other parameters the same) was very low, about 16 times weaker than the fitted wild-type value (S2 Fig).

## Fitting the model to an updated $\phi$X174 transcriptome

Since we were able to fit the model to coarse-grained qPCR data, we next fit the simulations to higher-resolution RNA-seq data. Logel and Jaschke [12] recently re-measured $\phi$X174 transcription using RNA-seq, and found several locations in the transcriptome where sharp changes in read abundance did not correspond to any known $\phi$X174 regulatory elements. To explain these discrepancies, they proposed one putative promoter and one putative terminator (called btss49 and RUT-3, respectively). The putative elements were computationally verified using promoter/terminator prediction tools but were not tested experimentally.

We were interested to see if simulations could provide evidence for or against the putative $\phi$X174 regulatory elements. To do so, we re-fit simulations of the canonical model (without putative regulatory elements) and the expanded regulatory model (with btss49 and RUT-3) to the updated RNA-seq measurements. After 4000 generations, all replicate simulations converged with similar mean RMSE values (S3 Fig). Fig 4 shows the parameter distributions from all fitted simulations. The mean estimated binding strength of btss49 is $1.7 \times 10^6 \text{M}^{-1}\text{s}^{-1}$, which is lower than the other three canonical promoter strengths (Fig 4A) but similar in magnitude to promoter B. The mean estimated termination efficiency for RUT-3 is 30%, similar to estimates for $T_F$ and $T_G$ (Fig 4B).

Interestingly, including two additional parameters (the putative regulatory elements) did not improve model fit (S3 Fig). In fact, the mean RMSE for the canonical model was lower than the mean RMSE for the expanded model. Also, parameter estimates for the other canonical regulatory elements were essentially unchanged in the fitted expanded model. These results suggest that the impact of btss49 and RUT-3 on $\phi$X174 transcription as a whole is minor. We also estimated a very low, almost negligible strength for btss49, suggesting that this may not be a true promoter, at least in the sense that it is probably not under strong selection to initiate

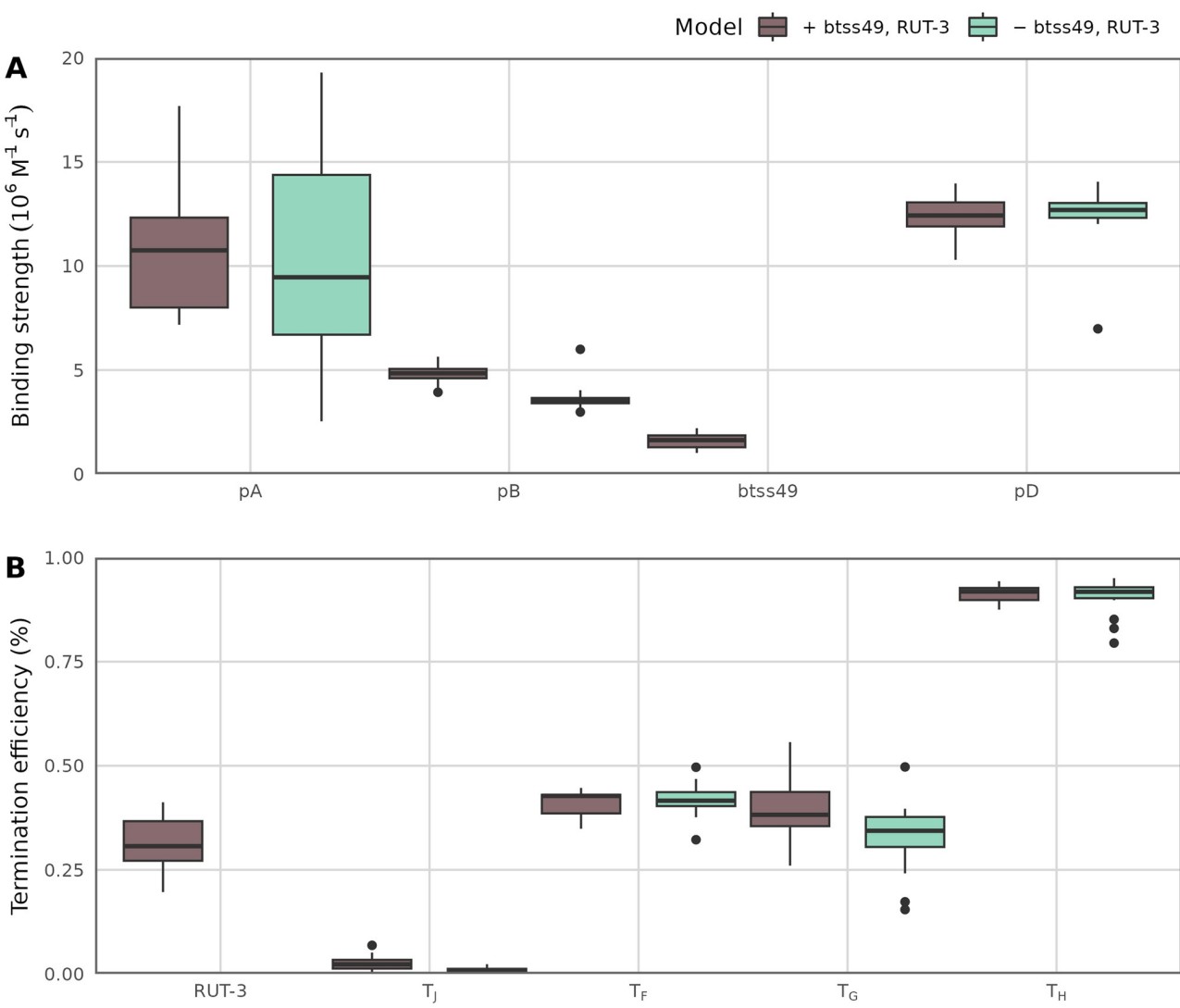

**Fig 4. Comparison of the canonical φX174 regulatory model to an expanded model with two putative regulatory elements.** Simulations were fit to RNA-seq data from [12]; promoter estimates (A) and terminator estimates (B) are shown for each model type (canonical model in brown, expanded/putative model in green). Here, each point represents a parameter from the best-fit simulation (the simulation with the lowest RMSE) for each of the 20 replicate simulations that was fit for each model type.

transcription. For the other element, a putative Rho-dependent terminator, we estimated a similar efficiency as the other weak canonical φX174 terminators.

For the canonical model (without putative regulatory elements), we note that the fit to RNA-seq data (Fig 4) is similar but not exactly identical to the fit to qPCR data (Fig 3). Most importantly, promoter pD is much stronger in the model fit to qPCR data than in the model fit to RNA-seq data. Also, terminators $T_J$ and $T_G$ are stronger in the fit to qPCR data. In fact, terminator $T_J$ is estimated to have near zero termination strength when fit to RNA-seq data. When comparing the measured gene expression levels (Fig 2 for qPCR vs. S4 Fig for RNA-seq), we see that expression differences are generally smaller for RNA-seq data than for qPCR data (for example, genes D, E, and J are expressed nearly 3× higher than genes B, K, C according to qPCR data but only about 2× higher according to RNA-seq data). These smaller

differences in measured gene expression levels naturally translate to weaker promoters and weaker terminators in the fitted model.

## Simulating decompressed φX174

In addition to promoter and terminator arrangements, another important aspect of φX174 genome organization is the presence of multiple overlapping genes. In a prior study, the φX174 genome was refactored to separate all primary coding sequences, in order to better understand the regulatory consequences of gene overlaps [5]. In this process (referred to as genome decompression) all partially or completely overlapping coding regions were copied and placed side-by-side. To prevent expression of the original gene copies (which would otherwise still be viable), initiation sites were disrupted by a series of point mutations made within each start codon. The decompressed φX174 strain was viable but showed reduced fitness compared to wild-type φX174, although the exact causes of the fitness reduction were unclear. Notably, protein A* abundances were significantly up-regulated, while protein B and C abundances were down-regulated [6] (Fig 5, blue bars). It is unclear whether gene order differences (which have been shown to affect gene expression and fitness), or unintended changes to regulatory sequences, or both, were responsible for altered φX174 gene expression.

We investigated some possible causes of disrupted φX174 protein production using a model for decompressed φX174 gene expression. We began by assuming that all promoter, terminator, and ribosome binding sites were unchanged during the decompression process (excluding deliberately knocked-out ribosome binding sites). We defined promoter and

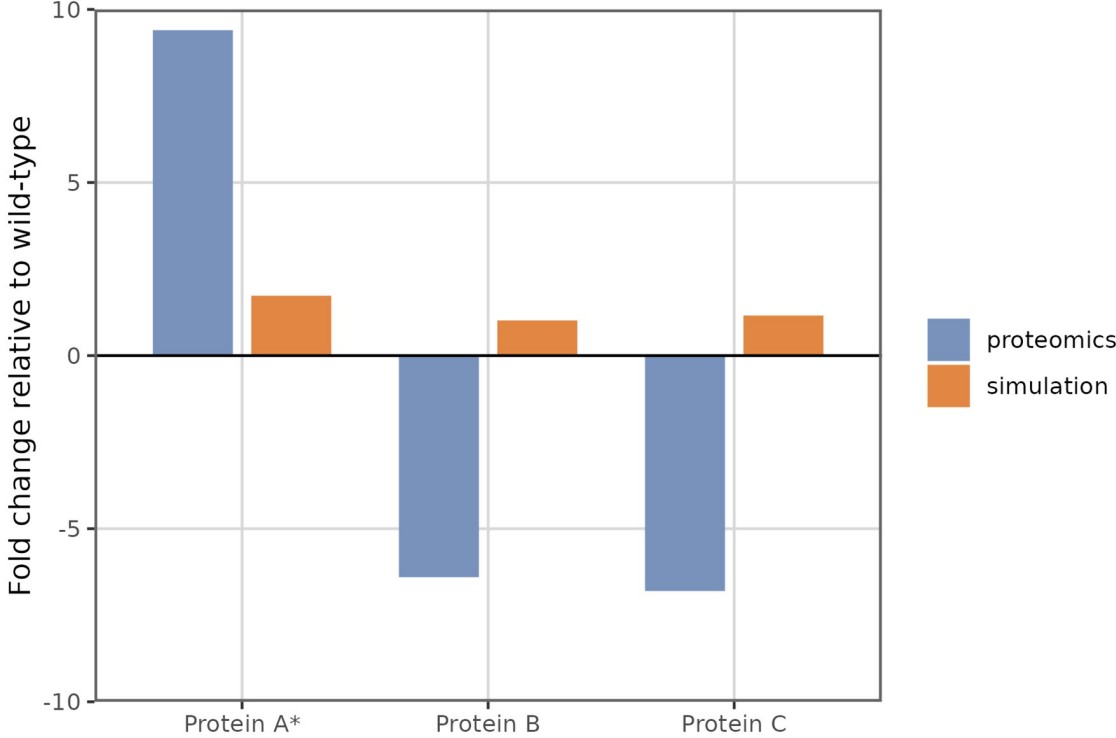

**Fig 5. Comparison of simulated and measured gene expression changes for a decompressed φX174 construct.** Measured protein fold-changes are in blue, and simulated fold-changes are in orange. The experimental data was collected by Wright et al. using targeted proteomics [6]. Only measurements for genes that were significantly over- or under-expressed relative to wild-type are shown.

terminator strengths using parameters from the wild-type model that was fit to qPCR data. By keeping gene regulatory elements consistent across the wild-type and decompressed architectures, we intended to isolate the effects of gene rearrangements. Fig 5 shows fold-changes in simulated decompressed protein abundances for the three genes with significant measured abundance differences. In the simulations, the impact of gene rearrangements on protein synthesis is very small (Fig 5, grey bars), suggesting that gene re-ordering/decompression alone is unlikely to account for the altered protein expression.

We reasoned that the decompression process may have also disrupted promoter or terminator sequences. We were interested if changing these parameters in our model could recapitulate the decompressed φX174 gene expression pattern. We fit parameters for promoters and terminators for the decompressed φX174 simulation to measured fold-changes in protein abundances (S5 Fig). When compared to mean parameter estimates from the fitted wild-type model, pD, $T_G$, and $T_H$ are much weaker in the decompressed model, while estimates for the other elements are mostly unchanged (S5A and S5B Fig). It is unlikely that $T_G$ and $T_H$ activities were directly affected by the decompression process, since they are outside of the region that was refactored. Although it is possible that decompression had an indirect effect on G and H expression, due to, for example, complex interplay between transcription and translation deregulation, our fit of transcription elements only could not have captured such an effect. In addition, the estimate for $T_G$ did not converge fully, unlike in the wild-type model. Also, it is impossible for the model to achieve a simultaneous increase in protein A/A* expression and decrease in B/C expression by only adjusting promoter and terminator strengths (S5C Fig). Overall, we were unable to achieve a good fit to the decompressed phenotype by only allowing promoters and terminators to vary. Taken together, these results point to a cause other than gene rearrangements or accidental disruption of promoter/terminator sequences for the altered protein expression phenotype observed for decompressed φX174.

## Discussion

We have developed a mechanistic computational simulation for bacteriophage φX174 gene expression, and we have used this model to estimate the relative activities of φX174 promoters and terminators by systematically fitting these parameters to transcription data. Our estimates are broadly consistent with empirical measurements for promoter and terminator strengths, but with some notable discrepancies, in particular for the promoters. We have used the same parameter estimation strategy to assess two putative regulatory elements. The alternative/putative regulatory model did not meaningfully improve the ability of the simulation to explain variations in φX174 transcript abundances, and estimates for canonical elements were essentially unchanged. We conclude from these findings that the evidence in favor of these putative elements is weak. Finally, we have used our model to simulate expression of a decompressed version of the φX174 genome in which overlapping genes have been placed side-by-side. We observe that our model does not recapitulate the observed expression changes from decompression, even after re-fitting promoter and terminator strengths. Our results suggest that the process of engineering the decompressed genome affected aspects of gene regulation other than transcription. Overall, our work helps to clarify aspects of transcription regulation in the important model organism φX174.

We fit the same computational model of φX174 to two different sources of bacteriophage expression data, a set of fairly coarse-grained qPCR measurements [13] and a much-more fine-grained set of RNA-seq measurements [12]. Overall the estimated promoter and terminator strengths were comparable, though we note that several regulatory elements were estimated to be weaker when the model was fit to RNA-seq data than when it was fit to qPCR data. These

weaker estimates for regulatory elements are consistent with the input gene expression data sets, which show consistently smaller differences in gene expression levels for the RNA-seq data than for the qPCR data. Unfortunately we have no way of assessing which estimates are more realistic. RNA-seq data can suffer from various biases [19, 20], though in direct comparisons to qPCR measurements the observed differences tend to be small [21]. Importantly, here the two data sets were also obtained by different groups using different protocols and materials, so that the observed differences in gene expression levels may also, at least in part, have been caused by these factors.

We also considered two different regulatory models of $\phi$X174, the canonical model widely used in the literature and an expanded model containing an additional promoter and an additional terminator [12]. We could fit the expanded model only to the RNA-seq data, as the qPCR data lacked sufficient gene-level resolution to be informative for the expanded model. (Not every gene was assayed for the qPCR data, and instead the assumption was made that gene expression levels could only change at the precise genomic locations where canonical promoters and terminators are present.) In a direct comparison of the two models fit to the same RNA-seq data, we found only weak evidence in favor of the expanded model. The estimated strengths of the canonical regulatory elements were largely unchanged in the expanded model, and the putative new promoter and terminator were weak. Moreover, the RMSE was not lower for the expanded model. We emphasize, however, that our approach cannot be used to rule out the existence of these elements. They may well exist and be weak. What our approach does show, however, is that these elements are not critically important to produce a model that can describe the observed gene expression levels.

The $\phi$X174 genome contains several overlapping genes. Such gene arrangements are commonly observed in viruses and bacteriophages [22], though the evolutionary pressures that lead to overlapping genes are not well understood. When bacteriophage genomes are engineered such that all gene overlaps are removed, a process generally referred to as "decompression," the resulting phages tend to display reduced fitness or other growth defects [6, 23, 24]. *A priori,* such effects may be caused by packaging defects if the enlarged genome no longer fits into the phage particle, by disrupted gene regulation—as the expression level of a given gene often results from the aggregate activity of multiple, staggered promoters—, or by the accidental disruption of unknown ORFs or regulatory elements during the engineering efforts. For $\phi$X174, it was observed that decompression led to significantly altered gene expression patterns [6]. Thus, we asked here whether we could recapitulate these findings in our simulations. However, we found that when we held activities for promoters, terminators, and ribosome binding sites fixed (changing only the positions of $\phi$X174 genes), simulated gene expression levels were largely unchanged between the regular and the decompressed $\phi$X174 genome. We were also unable to recapitulate gene expression changes by adjusting promoter and terminator strengths in the model. Specifically, it was impossible to simulate an increase in protein A* expression while also decreasing protein B and C expression, as was observed experimentally for the decompressed strain. Our results indicate that the causes of altered expression for the decompressed $\phi$X174 are likely more complex, for example, a result of a combination of changes to transcription and translation initiation.

Even though $\phi$X174 was one of the first organisms whose genome was fully sequenced and it remains today a widely studied bacteriophage, it is not commonly used for mathematical or computational modeling studies, unlike bacteriophages Q$\beta$ [25–28], $\lambda$ [29, 30], or T7 [15, 31–34] (see also Ref. [35] and references therein). Therefore, we cannot offer much in terms of comparison of our results to prior $\phi$X174 modeling work. We emphasize, however, that our approach here was not to develop a one-off, purpose-built models that can describe only one particular organism or biological system, as commonly seen in the field of biological

simulation. Instead, we leveraged the Pinetree simulator [14], which—even though originally developed for T7—is a general-purpose prokaryotic gene expression simulator that can be adapted relatively easily to any system of interest. All that is required to adapt Pinetree to a new organism is a detailed list of where in the genome individual genes start and end, and where promoters, terminators, and other regulatory elements are located. Consequently, any bacteriophage with a well annotated genome can be simulated in Pinetree with little effort. We note that the Pinetree approach was inspired by TABASCO [33], which similarly aimed to be a general-purpose simulator but which to our knowledge was never used for any application other than simulating bacteriophage T7.

## Materials and methods

To construct models for φX174, we used a customizable gene expression simulation framework called Pinetree [14], which uses an implementation of the Gillespie Stochastic Simulation Algorithm [36]. The Pinetree simulations take as input a target organism's genomic information, including the positions of genes, promotors, terminators, and ribosome binding sites. The user defines the number of cellular resources available at the start of a simulation, as well as any additional reactions involved in target gene expression or cell regulation. Pinetree was previously used to model bacteriophage T7's infection cycle [15]. Since T7 and φX174 infect the same host (*E.coli*), we began by downloading simulation materials (python scripts and control files) from the T7 project (https://github.com/benjaminjack/phage_simulation) and used these as a starting point for the φX174 model.

### Defining the φX174 genome

We downloaded a reference genome sequence for φX174 (NC_001422.1) from the National Center for Biotechnology Information (NCBI). We used a Python script to extract gene start and stop locations, and combined these with literature values for promoter and terminator locations [3, 4, 12] (Table 1). Since Pinetree requires linear genome sequences whereas the φX174 genome is circular, we linearized the genome in the following manner: First, we arbitrarily defined a location 63 nucleotides upstream of the gene A start site as location zero and cut the circular genome at this location. This cut location was chosen such that it did not overlap with any known genes or regulatory elements. The cut genome was then used as a linear genome in the Pinetree simulation. However, we also modified the Pinetree code such that any polymerases reaching the end of the linearized genome would not fall off but instead proceed to the beginning of the genome and continue transcription there. Furthermore, because Pinetree does not support certain genomic architectures, such as transcription elements that overlap with ribosome binding sites, we made minor adjustments to several genomic elements to accommodate these constraints. In doing so, we endeavored to keep the sizes and relative positions of all elements as close to original as possible. The revised element locations after linearization and removal of conflicts are provided in Table 1.

### Defining the host-cell environment

We simulated φX174 gene-expression dynamics for a single viral particle infecting a simplified version of an *E. coli* cell. The simulated host-cell environment contains resources needed by the phage (ribosomes and RNA polymerases), the majority of which are bound to host nucleic acids at the start of the simulation (Table 2). As the simulation progresses, polymerases and ribosomes dissociate from host DNA and RNA and become available to φX174. We modeled host-cell gene expression by simulating transcription and translation of a single, arbitrary *E. coli* gene of an average length. The rate constants for *E. coli* gene expression processes were

**Table 1. Locations of $\phi$X174 genome elements.** Here, "Reference" refers to the location of the elements in the reference genome sequence that was downloaded from NCBI. Coordinates are provided for simulations of the wild-type and decompressed strains.

| Element | Reference | Simulation (WT) | Simulation (decomp.) |
|---|---|---|---|
| Gene A | 3980..5386, 0..136 | 63..1605 | 64..1605 |
| Gene A* | 4496..5386, 0..136 | 579..1605 | 751..1605 |
| Gene B | 5074..5386, 0..51 | 1157..1520 | 1621..1983 |
| Gene K | 50..221 | 1519..1690 | 1998..2168 |
| Gene C | 132..393 | 1601..1862 | 2183..2443 |
| Gene D | 389..848 | 1858..2317 | 2459..2917 |
| Gene E | 567..843 | 2036..2312 | 2933..3208 |
| Gene J | 847..964 | 2316..2433 | 3225..3341 |
| Gene F | 1000..2284 | 2469..3753 | 3386..3753 |
| Gene G | 2394..2922 | 3863..4391 | 3864..4391 |
| Gene H | 2930..3917 | 4399..5386 | 4399..5385 |
| pA | 3927..3962 | 1..45 | 1..45 |
| pB | 4863..4899 | 880..925 | 818..863 |
| btss49 | 129..155 | 1598..1643 | – |
| pD | 320..358 | 1781..1826 | 2363..2408 |
| $T_H$ | 3871..4007 | 55..56 | 55..56 |
| RUT-3 | 5160..5239 | 1229..1230 | – |
| $T_J$ | 901..1091 | 2436..2437 | 3345..3346 |
| $T_F$ | 2250..2441 | 3796..3797 | 3796..3797 |
| $T_G$ | 2891..3111 | 4400..4401 | 4392..4393 |

defined using values from the prior T7 model [15] (Table 3). We held the total number of polymerases and ribosomes fixed, such that $\phi$X174 and *E. coli* needed to compete over a finite pool of resources for the duration of the simulation. With the parameters chosen, the steady-state number of polymerases and ribosomes occupied by $\phi$X174 was a small fraction of the total, less than 10%. We note that this fraction could be adjusted by tuning the rate constants for *E. coli* gene expression, which would increase (or decrease) phage gene expression rates uniformly across all genes, however, for simplicity we decided to keep the same host-cell parameters as the T7 model.

## Model optimization

We used a simple iterative optimization procedure to fit models of $\phi$X174 gene expression to transcription data. We initialized simulations with random values for the three canonical $\phi$X174 promoters and four terminators (or, alternatively for the non-canonical model, the

**Table 2. Molecular species in $\phi$X174 simulations.**

| Species | Description | Init. count |
|---|---|---|
| RNA polymerase | Free RNA polymerase | 0 |
| RNA polymerase bound to host DNA | Polymerase that is unavailable to $\phi$X174 | 1800 |
| Ribosome | Free ribosome | 30 |
| Ribosome bound to host RNA | Ribosome that is unavailable to $\phi$X174 | 10000 |
| *E. coli* genome | Fragment of host genome | 0 |
| *E. coli* transcript | Host transcript of average length | 0 |
| $\phi$X174 genome | Phage genome | 1 |

**Table 3. Host-cell reactions and rate constants in φX174 simulations.**

| Reaction | Rate constant |
|---|---|
| RNA polymerase + *E.coli* genome → Bound RNA polymerase | $10^7 \mathrm{M}^{-1}\mathrm{s}^{-1}$ |
| Bound RNA polymerase → RNA polymerase + *E.coli* genome + *E.coli* transcript | $0.04\mathrm{s}^{-1}$ |
| *E.coli* transcript + Ribosome → Bound ribosome | $10^6 \mathrm{M}^{-1}\mathrm{s}^{-1}$ |
| Bound ribosome → Ribosome + *E.coli* transcript | $0.04\mathrm{s}^{-1}$ |
| *E.coli* transcript → Degraded transcript | $1.925 \times 10^{-3}\mathrm{s}^{-1}$ |

seven elements plus one putative promoter and one putative terminator). The initial terminator values, which correspond to the efficiency of termination, were sampled from the interval (0, 1]. Promoter values were sampled from a normal distribution with mean 12 and standard deviation 3, and then scaled by a factor of $10^6$ to convert to units of $\mathrm{M}^{-1}\mathrm{s}^{-1}$ (for mesoscopic binding rates).

During each iteration of model training, a single promoter or terminator was randomly selected and adjusted. Promoter strengths were adjusted by multiplying by $2^n$, where $n$ is a random value drawn from a normal distribution with a mean of 0 and standard deviation of 0.1. Terminator strengths were adjusted by adding a random value drawn from a normal distribution with mean of 0 and standard deviation of 0.05. After adjusting the chosen parameter, we ran five simulations and used the averages from these runs to calculate the error between simulation output and training data.

Experimentally measured transcript abundances are usually reported as relative quantities, however, we found that using absolute quantities worked better for model fitting. To prepare experimental data for model optimization, we converted relative φX174 transcript abundances from literature to an absolute, discrete quantity for each gene. We set the target total simulated transcript quantity to 1500—that is, the transcript counts for all 11 φX174 genes should sum to around 1500 at the endpoint of the simulation. We found that when using these parameters, simulations take about 30 seconds each to run. Then, we scaled the amount for each individual gene according to reported relative abundances, generating a final data set of 11 values. This was done separately for qPCR and RNA-seq. Then, the mean squared error can be computed,

$$\mathrm{MSE}(y, \hat{y}) = \frac{\sum_{i=0}^{N-1} (y_i - \hat{y}_i)^2}{N}, \tag{1}$$

where $N$ is 11, the number of φX174 genes.

To fit the decompressed model, we used exactly the same procedure as described above, except the MSE was calculated for protein abundances instead of transcript abundances. For the optimization data, we started with simulated protein abundances from the wild-type model (specifically, the model that was fit to qPCR data) and scaled these according to the protein fold-changes reported in [6] to generate the final optimization data set.

## Supporting information

**S1 Table. Empirical measurements of φX174 promoter and terminator strengths.** For promoters pA and pB, the first/top value is the activity measured from PCR-generated fragments, and the second/bottom value is for promoters cloned into reporter plasmids. For promoter pD, activity could be measured for the PCR-generated fragments only. Measured promoter values are reported in [4], and measured terminator values are reported in [17]. (XLSX)

**S1 Fig. RMSE of models trained on qPCR data.** Each point corresponds to the simulation that had the best (lowest) RMSE over the 4000 training generations. The dashed line marks the manually-defined cutoff score used in downstream analysis.
(PNG)

**S2 Fig. Simulations of $\phi$X174 gene expression with the order of promoter A (pA) and terminator H (T$_H$) reversed.** Left panel: simulation of wild-type $\phi$X174 (pA before T$_H$) with parameter values obtained from fitting simulations to qPCR data. Middle panel: simulation with pA/T$_H$ order reversed (T$_H$ before pA) and the same pA binding strength as the fitted simulation (pA = 8.44). Transcript abundances for genes A/A* are about 10 times greater than wild-type, and abundances for genes B/K/C are about 2 times greater than wild-type. Right panel: simulation with pA/T$_H$ order reversed (T$_H$ before pA) and a pA binding strength of 0.5. The value of 0.5 was obtained by manually adjusting promoter A to revert transcript abundances for genes A/A*/B/K/C back to their wild-type ratios. For all panels, simulated transcript abundances are steady-state quantities averaged over five replicate simulations.
(PNG)

**S3 Fig. Distribution of RMSEs for models trained on RNA-seq data.** Each point corresponds to one simulation that had the best (lowest) RMSE over the 4000 training generations.
− btss49, RUT-3: Simulations with the canonical $\phi$X174 regulatory model. + btss49, RUT-3: Simulations with an alternative regulatory model proposed by Logel and Jaschke [12], with one additional promoter and one additional terminator. The mean RMSE for simulations with the alternative model is significantly higher than the mean for simulations with the canonical model only ($p < 0.05$, Student's t-test).
(PNG)

**S4 Fig. Simulated transcript abundances from $\phi$X174 models fit to RNA-seq data.** Left panel: Target $\phi$X174 transcription data, measured using RNA-seq. Transcript abundances shown are per million reads, normalized to gene A. The transcription data was collected by Logel and Jaschke [12]. Middle panel: Output from the best fit simulation (simulation with the lowest RMSE out of 20 independently fit models) with putative promoter btss49 and putative terminator RUT-3. Right panel: Output from the best fit simulation of the canonical regulatory model.
(PNG)

**S5 Fig. Fitting models of decompressed $\phi$X174 protein expression.** The target protein abundance was prepared by adjusting simulated wild-type protein abundances to match fold-change differences as reported by Wright et al. [6] (see also Materials and methods). Promoter distributions (A) and terminator distributions (B) from 20 independently fit models. In both A and B, the light blue diamonds are the mean parameter estimates from fitting the wild-type model to qPCR data. C: Final simulated protein abundances from the decompressed model. Simulated wild-type protein abundances (left panel) and the target, adjusted protein abundances (middle panel) are shown for reference.
(PNG)

## Author Contributions

**Conceptualization:** Alexis M. Hill, Claus O. Wilke.

**Data curation:** Alexis M. Hill.

**Formal analysis:** Alexis M. Hill, Tanvi A. Ingle.

**Funding acquisition:** Claus O. Wilke.

**Investigation:** Alexis M. Hill, Tanvi A. Ingle.

**Methodology:** Alexis M. Hill, Tanvi A. Ingle.

**Project administration:** Alexis M. Hill, Claus O. Wilke.

**Resources:** Alexis M. Hill, Tanvi A. Ingle.

**Software:** Alexis M. Hill, Tanvi A. Ingle.

**Supervision:** Alexis M. Hill, Claus O. Wilke.

**Validation:** Alexis M. Hill.

**Visualization:** Alexis M. Hill, Tanvi A. Ingle, Claus O. Wilke.

**Writing – original draft:** Alexis M. Hill, Tanvi A. Ingle, Claus O. Wilke.

**Writing – review & editing:** Alexis M. Hill, Tanvi A. Ingle, Claus O. Wilke.

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
