## [Decision Letter · Decision Letter 0]

22 Sep 2024

PONE-D-24-39861A computational model for bacteriophage ϕX174 gene expressionPLOS ONE

Dear Dr. Wilke,

Thank you for submitting your manuscript to PLOS ONE. After careful consideration, we feel that it has merit but does not fully meet PLOS ONE’s publication criteria as it currently stands. Therefore, we invite you to submit a revised version of the manuscript that addresses the points raised during the review process.

We look forward to receiving your revised manuscript.

Kind regards,

Sayed Haidar Abbas Raza

Academic Editor

PLOS ONE

“This work was supported by a National Institutes of Health grant R01 GM088344 and by the Jane and Roland Blumberg Centennial Professorship in Molecular Evolution and the Dwight W. and Blanche Faye Reeder Centennial Fellowship in Systematic and Evolutionary Biology at The University of Texas at Austin.”

“This work was supported by a National Institutes of Health grant R01 GM088344 and

by the Jane and Roland Blumberg Centennial Professorship in Molecular Evolution and

the Dwight W. and Blanche Faye Reeder Centennial Fellowship in Systematic and

Evolutionary Biology at The University of Texas at Austin.”

“This work was supported by a National Institutes of Health grant R01 GM088344 and by the Jane and Roland Blumberg Centennial Professorship in Molecular Evolution and the Dwight W. and Blanche Faye Reeder Centennial Fellowship in Systematic and Evolutionary Biology at The University of Texas at Austin.”

Reviewers' comments:

Reviewer's Responses to Questions

**Comments to the Author**

1. Is the manuscript technically sound, and do the data support the conclusions?

Reviewer #1: Yes

Reviewer #2: Yes

2. Has the statistical analysis been performed appropriately and rigorously? 

Reviewer #1: N/A

Reviewer #2: Yes

3. Have the authors made all data underlying the findings in their manuscript fully available?

Reviewer #1: Yes

Reviewer #2: Yes

4. Is the manuscript presented in an intelligible fashion and written in standard English?

Reviewer #1: Yes

Reviewer #2: No

5. Review Comments to the Author

Reviewer #1: The manuscript "A computational model for bacteriophage φX174 gene expression" talks about the construction of a computational model for φX174 and using the model to study gene regulation during the phage infection cycle. They build a model for bacteriophage φX174 gene expression using a customizable stochastic simulation framework that was previously used to model bacteriophage T7 infection dynamics.They modeled the phage genome as a linear molecule with a start site 63 nucleotides upstream of gene A. To simulate circular genome transcription, polymerases that reach the end of the genome without terminating automatically begin another round of transcription starting from the beginning of the genome. They estimated the strengths of promoters and terminators by fitting the simulations to φX174 transcription data. They first fit the model to qPCR data of φX174 and then they fit the simulations to higher-resolution RNA-seq data. For the canonical model (without putative regulatory elements), they found that the fit to RNA-seq data is similar but not exactly identical to the fit to qPCR data. The φX174 genome was previously refactored to remove all gene overlaps a process referred to as genome decompression. In the decompressed φX174 variant, all partially or completely overlapping coding regions were copied and placed side-by-side. The decompressed φX174 strain was viable but showed reduced fitness compared to wild-type φX174, although the exact causes of the fitness reduction were unclear. The authors conclude by saying that their work demonstrates that φX174 gene regulation is well described by the canonical set of promoters and terminators widely used in the literature. The authors also emphasize, that the approach here was not to develop a one-off, purpose-built models that can describe only one particular organism or biological system, as commonly seen in the field of biological simulation. Instead, they leveraged the Pinetree simulator, which is a general-purpose prokaryotic gene expression simulator that can be adapted relatively easily to any system of interest.

This manuscript is well written. The methods are explained in detail and the conclusions are based on the results. The manuscript though did not provide any new knowledge to the field but provides a framework for future studies.

Reviewer #2: This research article presents an impressive and detailed computational simulation of ϕX174 gene expression, offering significant insights into the bacteriophage’s transcriptional regulation. The use of mechanistic modeling to explore decompression and regulatory element behavior is well-executed, and the comparison between wild-type and decompressed models contributes valuable knowledge to the field of viral gene regulation. The use of two distinct data sources (qPCR and RNA-seq) strengthens the robustness of the findings.

Positive Aspects:

The study uses sophisticated computational simulations to understand viral gene regulation, which is a valuable contribution to virology research.

The comparison between wild-type and decompressed ϕX174 genomes offers new insights into the effect of gene rearrangement on transcription and expression.

The use of both qPCR and RNA-seq data adds depth to the findings, providing a comparative analysis of different regulatory models.

Suggestions for Minor Corrections:

Grammar and Spelling:

Line 185: Change "may also have disrupted" to "may have also disrupted."

Line 203: "We have used this model" should be "We used this model" for consistency with past tense.

Line 216: "engineering the decompressed genome had effects" could be improved to "engineering of the decompressed genome affected."

Line 254: "the expression level of a given gene is often the aggregate result" should be "often results from the aggregate."

Technical Consistency:

The reasoning for why a substantial decrease in TG and TH activities is unlikely should be elaborated for clarity, as it jumps to a conclusion without adequate evidence.

In the Results section, the explanation of model fitting for decompressed vs. wild-type ϕX174 is detailed but could benefit from a clearer breakdown of specific observations with more explanation on fitting difficulties (such as TG variability).

Flow and Clarity:

The overall flow between sections can be improved. In particular, the transition between the discussion of decompression and transcription regulation feels abrupt. More context on decompression's impact on transcription could help bridge these ideas.

These corrections and improvements can strengthen both the clarity and technical presentation of this valuable research.

6. PLOS authors have the option to publish the peer review history of their article (what does this mean?). If published, this will include your full peer review and any attached files.

Reviewer #1: No

Reviewer #2: **Yes: **Simna Saraswathi Prasannakumari

---

## [Author Response · Author response to Decision Letter 0]

14 Oct 2024

A detailed response to the reviewers has been provided as a separate file. We have also uploaded a version of the manuscript with revisions highlighted.

---

## [Editor Report · Decision Letter 1]

17 Oct 2024

A computational model for bacteriophage ϕX174 gene expression

PONE-D-24-39861R1

Dear Dr.Claus O. Wilke,

We’re pleased to inform you that your manuscript has been judged scientifically suitable for publication and will be formally accepted for publication once it meets all outstanding technical requirements.

Kind regards,

Sayed Haidar Abbas Raza

Academic Editor

PLOS ONE

Additional Editor Comments (optional):

Thanks to response the comments
---

## [Editor Report · Acceptance letter]

21 Oct 2024

PONE-D-24-39861R1 

PLOS ONE

Dear Dr. Wilke, 

I'm pleased to inform you that your manuscript has been deemed suitable for publication in PLOS ONE. Congratulations! Your manuscript is now being handed over to our production team.

Kind regards, 

on behalf of

Dr. Sayed Haidar Abbas Raza 

Academic Editor

PLOS ONE